# The Effect of Atmospheric Dielectric Barrier Discharge Cold Plasma Treatment on the Nutritional and Physicochemical Characteristics of Various Legumes

**DOI:** 10.3390/foods12173260

**Published:** 2023-08-30

**Authors:** Yingmei Wu, Xuewei Feng, Yingying Zhu, Shiyu Li, Yichen Hu, Yang Yao, Nong Zhou

**Affiliations:** 1Chongqing Engineering Laboratory of Green Planting and Deep Processing of Famous-Region Drug in the Three Gorges Reservoir Region, College of Biology and Food Engineering, Chongqing Three Gorges University, Chongqing 404120, China; wuyingmei0927@126.com; 2Henan Key Laboratory of Cold Chain Food Quality and Safety Control, College of Food and Bioengineering, Zhengzhou University of Light Industry, Zhengzhou 450001, China; fxw15517546372@hotmail.com (X.F.); zhuying881020@163.com (Y.Z.); 3Institute of Crop Science, Chinese Academy of Agricultural Sciences, Beijing 100081, China; lsy2977831386@hotmail.com; 4Sichuan Engineering & Technology Research Center of Coarse Cereal Industrialization, Key Laboratory of Coarse Cereal Processing, Ministry of Agriculture and Rural Affairs, School of Food and Biological Engineering, Chengdu University, Chengdu 610106, China; huyichen0323@126.com

**Keywords:** legumes, lipoxygenase, DBD-ACP, physicochemical structure, digestion properties

## Abstract

High activity of lipoxygenase (LOX) has been identified as a primary cause of oxidative rancidity in legumes. In this study, the application of dielectric barrier discharge atmospheric cold plasma (DBD-ACP) (5 W, 10 min) resulted in an obvious decrease in LOX activity in mung bean (MB), kidney bean (KB), and adzuki bean (AB) flours by 36.96%, 32.49%, and 28.57%, respectively. Moreover, DBD-ACP induced significant increases (*p* < 0.05) in content of soluble dietary fiber, saturated fatty acids, and methionine. The starch digestibility of legumes was changed, evidenced by increased (*p* < 0.05) slowly digestible starch and rapidly digestible starch, while resistant starch decreased. Furthermore, DBD-ACP treatment significantly affected (*p* < 0.05) the hydration and thermal characteristics of legume flours, evidenced by the increased water absorption index (WAI) and gelatinization temperature, and the decreased swelling power (SP) and gelatinization enthalpy (ΔH). Microscopic observations confirmed that DBD-ACP treatment caused particle aggregation.

## 1. Introduction

Legumes play a significant role in sustainable food production, and regular consumption of legumes has been associated with various health benefits including lowering blood cholesterol levels, improving blood glucose disorder, and reducing the risk of heart disease [1,2]. Hence, legume-based foods are becoming increasingly popular among health-conscious consumers and are also an important food source for vegetarians and vegans. Mung bean (MB), kidney bean (KB), and adzuki bean (AB) are the main types of minor legumes in China. The China Rural Statistical Yearbook reports MB’s annual domestic output at approximately 50.8 million tons, with AB yielding around 20.8 million tons in 2020. Additionally, an alternate study indicates KB’s annual domestic production to be roughly 11 million tons [3]. Three legumes occupy an important place in the development of the food industry.

The high activity of lipoxygenase (LOX) in legumes results in the development of undesirable off-flavors during storage [4]. In industrial production, thermal treatments are commonly employed to inactivate LOX in crops. However, these treatments are associated with low thermal efficiency, high energy consumption, and nutrient loss [5]. Atmospheric cold plasma (ACP) technology, a non-thermal processing method, generates a plethora of reactive particles, including reactive oxygen and nitrogen species, free radicals, ultraviolet radiation, and charged particles, through high-voltage discharges at ambient temperatures [6]. This technology finds application in non-thermal sterilization, enzyme inactivation, mycotoxin degradation, and modification of food constituents within the food industry [7]. One extensively explored approach to ACP is the dielectric barrier discharge (DBD) method, notable for its safety and wide adoption among cold plasma techniques. The reduction of inherent enzyme activities such as polyphenol oxidase, peroxidase, LOX, lipase, and lactate dehydrogenase through DBD-ACP treatment has been widely documented, particularly in regard to LOX [8]. It has been reported that DBD-ACP treatment (24 kV, 25 min) decreased the LOX activity of wheat germ by 49.98% [9]. Additionally, another study demonstrated that after subjecting Mizhi millet to DBD-ACP treatment (25 kV, 12 min), LOX activity decreased by 57.5% [10]. Moreover, DBD-ACP treatment has also been applied to improve the edible and processing quality of crops. It was demonstrated that the water absorption index (WAI) and water solubility index (WSI) of parboiled rice flour increased after DBD-ACP treatment (30–50 W, 5–15 min), along with a rise in the gelatinization temperature, indicating an improvement in the cooking quality of parboiled rice flour [11]. Similarly, DBD-ACP treatment (20 kV, 30 s) has been found to increase the solubility and decrease the swelling power (SP) of cereal starches [12]. However, few studies have focused on the inactivation of LOX in legumes through DBD-ACP treatment, as well as the subsequent impact of DBD-ACP treatment on the processing and nutritional characteristics of legumes.

In this study, the impact of DBD-ACP treatment on the activity of LOX in MB, KB, and AB flours was first assessed, and then the influences of DBD-ACP treatment on the nutritional and digestive characteristics of these legumes were examined. At last, the alterations in physicochemical and microstructural properties induced by DBD-ACP treatment were investigated.

## 2. Materials and Methods

### 2.1. Raw Materials

Seeds of mung bean (MB, Zhonglyu 8), kidney bean (KB, Weining white kidney bean), and adzuki bean (AB, Jihong 352) were provided by the Chinese Academy of Agriculture Sciences (Beijing, China). Porcine pancreatic α-amylase (EC 3.2.1.1), amyloglucosidase (EC 3.2.1.3), fatty acid standards, LOX standard, Tween-20, and linoleic acid were procured from Shanghai Yuanye Biotechnology Co., Ltd. (Shanghai, China). Amino acid standards and glucose standards were sourced from Macklin Company (Shanghai, China). Other chemical reagents were of analytical grade and were obtained from BOHUA Chemical Reagent Biotechnology Co., Ltd. (Tianjin, China).

### 2.2. Preparation of Samples

The MB seeds (9.4% moisture content), KB seeds (11.1% moisture content), and AB seeds (10.6% moisture content) were finely ground and were sieved using a 60-mesh filter to yield whole bean flour. For this study, a DBD-ACP instrument (CTP-2000K, Nanjing Suman Electronic Co., Ltd., Nanjing, China) was utilized, consisting of an aluminum sample tray, a power supply, two circular aluminum plates functioning as electrodes and ceramic materials forming a dielectric barrier. The reactor dish was made of quartz with dimensions of 70 mm (outer diameter), 50 mm (inner diameter), 4 mm (depth), and 8 mm (wall thickness). Plasma was generated between the aluminum plate electrodes, as depicted in Figure 1. The sample treatment was conducted at atmospheric pressure, employing argon as the working gas. Each sample (3 g) was uniformly spread over the aluminum sample tray at room temperature with a load of 1.53 kg/m^2^, and the electrode spacing was maintained at 5 mm. The input power was fixed at 5 W and the exposure time varied from 0 to 20 min.

### 2.3. Determination of LOX Activity

The LOX activity was assessed as previously described by Niu et al. with some refinements [13]. An amount of 20 mg of Tween-20 was combined with 700 mg of linoleic acid in 5 mL of water, and the mixture was subjected to 15 min of ultrasonic agitation. Afterward, 5.5 mL of NaOH solution (0.5 mol/L) was added, and distilled water was used to adjust the final volume to 25 mL. One gram of legume flour was combined with 25 mL PBS (0.05 mol/L, pH 7) and underwent oscillation extraction (4 °C, 15 min), then the crude enzyme extract was obtained. After centrifugation (8000 rpm, 5 min) of the mixture, the supernatant was collected and stored (4 °C). For the measurement of LOX activity, the enzyme solution (0.2 mL) was thoroughly mixed with the substrate solution (2.8 mL). In the control group, the mixture of substrate solution and inactivated enzyme extract was monitored at 234 nm using an automatic enzyme marker (JC-1086A, Qingdao Juchuang Huanye Analytical Instrument Co., Ltd., Qingdao, China). The change in optical density (OD) value within 2 min was recorded. One unit of enzyme activity was defined as the amount of enzyme required to produce a change in OD value of 0.01 per minute.
(1)$$X(U\cdot {min}^{-1}\cdot g^{-1})=\frac{\Delta 234\times V_{2}\times V_{0}}{\Delta t\times 0.01\times V_{1}\times M}$$

In the equation above, X represents the LOX activity, Δt denotes the total reaction time in minutes, Δ234 indicates the change in absorbance at 234 nm during Δt, V_0_ represents the total volume of the reaction system in milliliters, V_1_ is the amount of crude enzyme extract added in milliliters, V_2_ signifies the total volume of the crude enzyme solution in milliliters and M represents the dry weight of the sample in grams.

### 2.4. Nutritional Components

The protein contents were determined using the Kjeldahl Method (AOAC, 950.09). The fat content was determined using Soxhlet extraction method (AOAC, 963.15). The 3,5-Dinitrosalicylic acid (DNS) method was employed to determine the carbohydrate content. A dietary fiber analysis kit (Beijing Zhimicro Technology Co., Ltd., Beijing, China) was used to analyze the content of soluble dietary fiber (SDF), insoluble dietary fiber (IDF), and total dietary fiber (TDF). All the obtained results were reported on the basis of dry matter.

### 2.5. Analysis of Amino Acids and Fatty Acids Composition

Quantitative amino acids analysis was performed using a fully automated amino acid analyzer (S433D, Sykam Scientific Instrument Co., Ltd., Munich, Germany), according to the method reported by Lisiewska et al. [14]. To produce alkaline hydrolysate, the samples were subjected to hydrolysis in 6 mol/L HCl. Determination of fatty acid composition was performed according to a study by Saeed Alkaltham et al. [15]. Using an Agilent 7890 analytical gas chromatography system (Agilent Technologies Co., Ltd., Palo Alto, CA, USA). The system was equipped with aflame-ionization detector, HP-5 column (30 m × 0.25 mm, 0.5 μm), and an autosampler was employed.

### 2.6. In Vitro Digestion of Starch

The in vitro starch digestibility was measured following a previously reported method by Englyst et al. with some modifications [16]. Briefly, different flours (500 mg) were, respectively, incubated with acetate buffer (25 mL, 0.1 M, pH 5.1) for 15 min. Next, porcine pancreatic α-amylase (290 U/mL, 4 mL) and amyloglucosidase (60 U/mL, 1 mL) were added with continuous shaking (37 °C) for 120 min. The reduced sugar concentration in the digestive juice was determined using DNS method at 0, 5, 10, 20, 30, 60, 90, and 120 min, respectively. The rapidly digestible starch (RDS), slowly digestible starch (SDS), and resistant starch (RS) were calculated using Equations (2)–(4) as follows:(2)$$RDS(\%)=(G_{20}-FG)\times 0.9\times 100$$
(3)$$SDS(\%)=(G_{120}-G_{20})\times 0.9\times 100$$
(4)$$RS\left(\%\right)=\left(TG-FG\right)\times 0.9\times 100-(RDS+SDS)$$

G_20_ and G_120_ represent the glucose contents produced by hydrolysis in 20 and 120 min, respectively. FG represents free glucose and TG represents total glucose.

### 2.7. Hydration Characteristics

The WAI, WSI, and SP were measured according to the method reported by Rosell et al. [17]. Firstly, legume flours (100 mg) were mixed with deionized water (20 mL) and incubated at 37 °C for 30 min with continuous stirring. After centrifugation (9860 rmp, 5 min), the supernatant and sediment were collected, respectively, and the supernatant was dried at 105 °C to obtain the dry matter weight of supernatant (Ws). The sediment was weighed to obtain the weight of wet sediment (Ww). The WAI, WSI, and SP were calculated using Equations (5)–(7) as follows:(5)$$\mathrm{WAI}\;(g/g)=\frac{Ww}{Wd}$$
(6)$$WSI(\%)=\frac{Ws}{Wd}\times 100$$
(7)$$SP(g/g)=\frac{Ww}{Wd\times (100-WSI)}$$

Wd represents the dry matter weight of the sample.

### 2.8. Thermal Property

A differential scanning calorimeter (DSC, Q100 TA Instruments, Newcastle, DE, USA) was employed to evaluate the thermal property. Three milligrams of the legume flour and 6 mL of distilled water were precisely weighed into a sealed aluminum pan, respectively. Then, after 24 h of standing, the sample pans were subjected to heating from 20 to 120 °C at a rate of 10 °C/min. The empty pan was utilized as a reference.

### 2.9. Fourier Transform Infrared Spectroscopy (FT-IR) Analysis

FT-IR spectroscopy was conducted using a spectrometer (VERTEX 70, Bruker Corporation, Karlsruhe, Germany) with a resolution of 4 cm^−1^ and 64 scans. The wavenumber range examined was from 4000 to 400 cm^−1^.

### 2.10. Scanning Electron Microscope (SEM)

The morphology of legume flours was observed using a scanning electron microscope (JSM-6490LV, Thermo Fisher Technology Co., Ltd., Shanghai, China) operating at an acceleration voltage of 20 kV.

### 2.11. Statistical Analysis

All experiments were conducted in triplicate, and data were presented as means ± standard deviations (SD). The differences in data were analyzed using Duncan test with a significance level of *p* < 0.05 in SPSS 26.0 (SPSS Inc., Chicago, IL, USA).

## 3. Results and Discussion

### 3.1. Effects of DBD-ACP Treatment on LOX Activity of Legume Flours

LOX is an oxidoreductive enzyme commonly found in legumes, which contributes to oxidative rancidity and unpleasant odors during storage. As depicted in Figure 2, the initial LOX activity in MB, KB, and AB was determined to be 862.50 U·min^−1^·g^−1^, 1731.25 U·min^−1^·g^−1^, and 1049.67 U·min^−1^·g^−1^, respectively. Following treatment with DBD-ACP, the LOX activity in all three legumes exhibited a significant reduction within the 0–10 min timeframe (*p* < 0.05). Specifically, after 10 min of treatment, the LOX activity in MB, KB, and AB decreased by 36.96%, 32.49%, and 28.57%, respectively. No further significant deactivation of LOX was observed with the extension of treatment time (*p* > 0.05). These findings indicate that DBD-ACP treatment is an effective method for inactivating LOX in MB, KB, and AB, with the optimal treatment conditions determined to be 5 W and 10 min. Essentially, LOX represents a protein entity. Previous studies have reported that one possible mechanism of reaction between plasma-generated reactive species and protein, such as the loss of α-helix structure and the change in some amino acid side chains, results in the loss of enzyme activity [18,19]. Similar results were reported by Wang et al. [10], wherein DBD-ACP treatment (15 kV, 12 min) resulted in a 48% reduction in LOX activity in foxtail millet.

### 3.2. Nutritional Composition

In this study, the main nutritional indicators of both treated (5 W, 10 min) and untreated MB, KB, and AB were determined, and the results were presented in Table 1. DBD-ACP treatment showed no significant effect on the total carbohydrate, protein, and fat content of the three legumes (*p* > 0.05), which was consistent with a previous study that reported DBD-ACP treatment had no obvious effect on the basic nutritional component in brown rice and basmati rice [20,21]. However, it is worth noting that some changes were observed in the contents of TDF, IDF, and SDF after DBD-ACP treatment (*p* < 0.05). The TDF content exhibited a slight increase of 6.1%, 1.5%, and 5.2% in MB, KB, and AB, respectively. Notably, a significant decrease (*p* < 0.05) in IDF content was observed in KB. Conversely, the SDF content showed a noticeable increase of 86.0%, 74.2%, and 31.8% in MB, KB, and AB, respectively. These alterations might be attributed to the breaking down of the cell walls induced by DBD-ACP, facilitating the release of SDF. Additionally, DBD-ACP might cause the breakdown of some insoluble polymers into shorter chain fragments, which improved solubility due to the increased exposure of polar functional groups, thus leading to an increase in the SDF portion [22,23].

### 3.3. Amino Acids Composition

As presented in Table 2, the results demonstrated that DBD-ACP treatment had a more pronounced effect on the amino acid composition of MB compared to KB and AB. Specifically, it significantly increased the levels of Val, Lle, Leu, and Phe in MB by 19.8%, 19.8%, 2.4%, and 7.3%, respectively. These changes can be attributed to the alteration of electron-rich groups within amino acids by the diverse reactive species generated during DBD-ACP treatment [24]. However, the Met levels in MB were reduced by 25%, while KB showed an increase of 37.5% and AB exhibited a 50% increase (*p* < 0.05). Pal et al. reported a reduction in methionine content for nonthermal plasma-treated short-grain rice flour, while the opposite effect was observed for long-grain rice flour [25]. It is well-known that although legumes possess a comprehensive amino acids composition, the deficiency of Met hinders its absorption within the human body, subsequently limiting the full utilization of other amino acids [26]. Thus, the DBD-ACP treatment could serve as one of the approaches to address the deficiency of methionine in legumes, thereby enhancing their nutritional value.

### 3.4. Fatty Acids Composition

The fatty acid compositions, including total saturated fatty acids (SFA), total monounsaturated fatty acids (MFA), and total polyunsaturated fatty acids (PFA), were presented in Table 3. In the untreated groups, the principal fatty acids were detected as C16:0, C18:2, and C18:3, which collectively accounted for 77.88% (untreated MB), 80.65% (untreated KB), and 62.20% (untreated AB) of the total fatty acids, respectively. These findings align with several previous studies, which reported that C16:0, C18:2, and C18:3 constituted 66.25% (MB), 82.4% (KB), and 91.6% (AB) of the total fatty acids [27,28,29]. In addition, following DBD-ACP treatment, there emerged distinct alterations in lipid composition. Specifically, the content of C16:0 experienced a significant rise of 33.0% (MB), 14.6% (KB), and 8.5% (AB), respectively. Conversely, the content of C18:2 exhibited marked declines of 44.4% (MB), 24.7% (KB), and 3.8% (AB), respectively. Meanwhile, the content of C18:3 displayed a 16.9% increase (MB) while demonstrating decreases of 22.5% (KB) and 14.5% (AB), respectively. A negative correlation emerged between the content of C16:0 and the soluble amino acid levels, whereas a positive correlation linked the content of C18:2 and C18:3 with soluble amino acids. These correlations were indicative of the oxidative changes in soluble amino acids induced by DBD-ACP treatment, consequently influencing the fatty acid constitution [30]. Following DBD-ACP treatment, a significant (*p* < 0.05) increase in total SFA and MFA content, along with a decrease in total PFA content, were observed across all three legumes. These findings were consistent with previously reported data. Silveira et al. also noted an increase in C16:0 content and a decrease in MFA content in guava-flavored whey beverages after cold plasma treatment [31]. This phenomenon might be attributed to the differing stability of chemical bonds, as the energy required to break bonds between carbon–carbon chains is higher than that needed for double bonds. Consequently, MFA, which possessed more double bonds, is more influenced by DBD-ACP treatment [32].

### 3.5. In Vitro Digestion of Starch

The contents of RDS, SDS, and RS as determined by in vitro simulated digestion were presented in Table 4. Following DBD-ACP treatment, the RDS and SDS contents of MB increased by 20.55% and 7.24%, respectively, while the RS content decreased by 6.20%. For KB, the RDS and SDS contents increased by 195.05% and 60.49%, respectively, and the RS content decreased by 16.48%. Similarly, in AB, the RDS and SDS contents increased by 12.62% and 2.32%, respectively, whereas the RS content decreased by 2.38%. The findings concurred with the investigation conducted by S. Gao et al., where they posited that DBD-ACP treatment engenders the conversion of electrical energy into thermal energy, thereby inducing partial gelatinization of starch. This phenomenon disrupts both the surface and internal structure of starch particles, resulting in an augmentation of enzyme binding sites [12]. On the other hand, etching due to DBD-ACP treatment is the main cause of cracks and pores on the surface of starch granules, which makes the starch molecules easier to cut into smaller fragments, which are more susceptible to enzymatic degradation during digestion [33]. Moreover, DBD-ACP treatment yields varying degrees of starch depolymerization, yielding products like maltose, maltotriose, and maltotetraose ultimately amplifying starch hydrolysis [12]. It was shown that these phenomena were positively correlated with treatment time and voltage [33]. It was worth noting that the changes observed in RDS, SDS, and RS in KB were highly significant. We speculated that this effect might be attributed to the partial inactivation of α-amylase inhibitor in KB caused by DBD-ACP treatment, resulting in improved starch digestibility. However, further research was required to corroborate this.

### 3.6. Hydration Properties

Hydration properties, including WAI, WSI, and SP, are commonly used indicators to evaluate the degree of protein denaturation and starch gelatinization. Figure 3 presented the WAI, WSI, and SP values for three legumes both with and without DBD-ACP treatment. The untreated samples exhibited WAI values of 4.18 g/g, 29.23% WSI, and 5.91 g/g SP for MB. Similarly, KB had WAI, WSI, and SP values of 5.4 g/g, 25.97%, and 7.29 g/g, respectively, while AB exhibited values of 4.75 g/g WAI, 20.07% WSI, and 5.94 g/g SP. These values aligned well with previous reports by Du et al. on whole bean flour [34]. Following DBD-ACP treatment, the WAI of treated MB, KB, and AB increased by 4.76%, 20.81%, and 4.40%, respectively, which supported the results of Thirumdas et al. who demonstrated that DBD could enhance the WAI of basmati rice flour with increased processing time and power [21]. Additionally, the WSI of treated MB, KB, and AB showed significant (*p* < 0.05) reductions of 35.19%, 34.86%, and 38.06%, respectively, after DBD-ACP treatment. In contrast, Chaple et al. reported different results regarding the effect of DBD-ACP treatment on WSI for wheat grain and wheat flour, where no significant differences were observed [35]. This discrepancy could be attributed to differences in sample matrix composition and the accessibility of starch to plasma treatment. Furthermore, the SP of treated MB, KB, and AB was significantly decreased by 18.60%, 2.54%, and 11.74%, respectively, after DBD-ACP treatment. A similar reduction in SP was reported for various starches treated with DBD [12].

### 3.7. Thermal Properties

The onset temperature (To), peak temperature (Tp), concluding temperature (Tc), and enthalpy (ΔH) values of samples are shown in Table 5. The determinations of To (80.86 °C, 74.31 °C, 69.10 °C), Tp (89.91 °C, 81.38 °C, 78.23 °C), Tc (93.32 °C, 86.84 °C, 82.65 °C), and ΔH (1.32 J/g, 2.14 J/g, 2.35 J/g) for untreated MB, KB, and AB were similar to previous results by Li et al. [36]. After DBD-ACP treatment, the To values of MB, KB, and AB increased by 4.72%, 0.89%, and 1.26%, respectively (*p* < 0.05). The increase in To might be attributed to the structural changes arising from the interaction between amylose molecules and amylose with lipids [37]. Similar observations of increased Tp in whole wheat grain and flour resulting from lower plasma power treatments had also been reported [35]. Furthermore, it has been reported that employing lower plasma power results in an elevated Tp value, whereas higher plasma power yields a contrasting pattern. This observation implies that distinct plasma processing conditions, such as treatment duration, intensity, and applied voltage, could give rise to fluctuations in Tp [38]. The DBD-ACP treatment led to a slight decrease in ΔH values for MB, KB, and AB by 3.03%, 22.43%, and 2.98%, respectively. Similar reductions in gelatinization enthalpy were reported in plasma-treated whole wheat grain and flour by Chaple et al., suggesting that cold plasma treatment requires less energy for the gelatinization process [35].

### 3.8. FT-IR Spectroscopy

The spectral features of MB, KB, and AB with and without DBD-ACP treatment were analyzed using FT-IR analysis. As depicted in Figure 4, the infrared spectrum peak positions of the three types of legumes were closely aligned, indicating a high similarity in their structure and functional groups. The absorption peak observed at 3425 cm^−1^ could be attributed to hydroxyl groups [12]. The bands observed at 2945 cm^−1^ and 1650 cm^−1^ were a result of the stretching of C-H, including CH and CH_2_ bending vibrations [39]. Additionally, the double absorption peaks near 1151 cm^−1^ and 1055 cm^−1^ corresponded to the stretching vibration peaks of the polysaccharide CO bond [40]. Following DBD-ACP treatment, the positions of the absorption peaks in the three legume groups remained unchanged, indicating that the plasma treatment did not generate new chemical groups. A similar finding was reported by S. Gao et al. for plasma-treated starch and native starch [12].

### 3.9. SEM

The morphology of MB, KB, and AB, both with and without DBD-ACP treatment, was examined using SEM (Figure 5). The micrographs revealed that the legume flour granules exhibited an irregular shape, ranging from polyhedral to ovoid, with smaller particles adhering to them, which consisted of protein and fiber fragments. Upon treatment, it was observed that smaller granules tended to aggregate in clusters, particularly in MB and KB flour. This finding was consistent with the results reported by S. Gao et al. who observed the aggregation of small-sized quinoa starch granules following DBD plasma treatment [12]. Importantly, no evident cracks or holes were observed in the legume flour granules after treatment. This outcome could be related to the differences in DBD treatment time and power, as supported by the findings of Thirumdas et al. [41]. They reported that no holes were observed in starch granules following 40 W-10 min plasma treatment, whereas, under the 60 W-10 min treatment condition, fissures were noticeable.

## 4. Conclusions

In this study, DBD-ACP treatment was found to be an effective way for inactivating LOX activity in legumes. Additionally, DBD-ACP treatment resulted in changes to the nutritional properties of legumes by modifying the profiles of dietary fiber, amino acids, and fatty acids. The digestibility of legumes was also influenced, with a noticeable increase in SDS being observed. Furthermore, DBD-ACP treatment had a significant impact on the hydration and thermal characteristics of legume flours, as it increased the WAI and gelatinization temperature while decreasing the SP and ΔH. Microscopic observations revealed that DBD-ACP treatment caused particle aggregation in legume flours. In conclusion, DBD-ACP treatment can be recommended for modifying the original storage and processing characteristics of legumes to facilitate their industrial applications. Regulatory approvals in the U.S. have been granted for a novel DBD direct plasma treatment targeting whole wheat grain, asserting its effectiveness in reducing spoilage. Moreover, the enhancement of desired functional attributes in foods via DBD-ACP treatment is an area of ongoing development. However, the majority of existing studies have been confined to laboratory scales. Transitioning towards pilot or industrial scales for practical implementation necessitates comprehensive documentation of plasma configuration, process technicalities, and material condition and attrition. Given the potential adverse consequences and limitations on the nutritional functional characteristics of foods, evaluating product safety and public health risks becomes imperative, thereby contributing to a more robust and secure production approach within the food industry.

## Figures and Tables

**Figure 1 foods-12-03260-f001:**
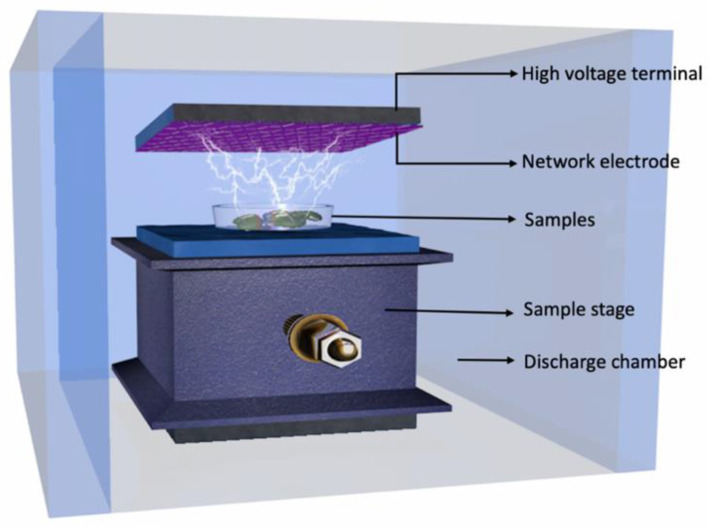
Schematic experimental setup of dielectric barrier discharge atmospheric cold plasma system.

**Figure 2 foods-12-03260-f002:**
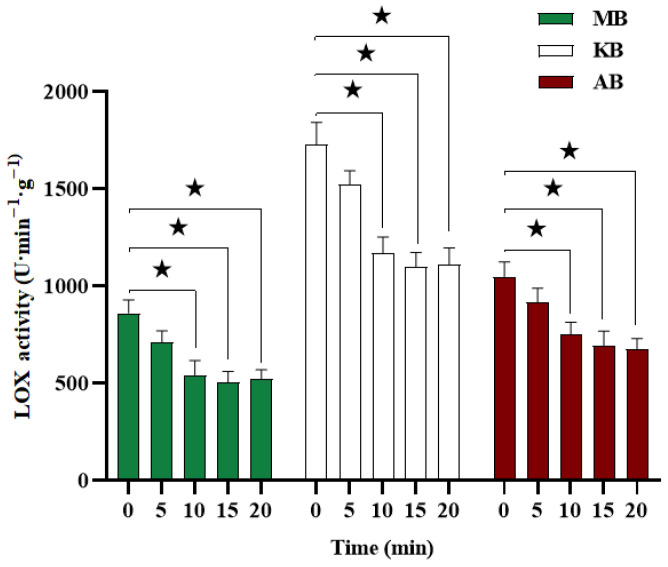
Effects of DBD-ACP treatment on LOX activity of different legumes. LOX means lipoxygenase, MB means mung bean, KB means kidney bean, and AB means adzuki bean. ★ In the same row of same kind of beans indicate significant differences at *p* < 0.05.

**Figure 3 foods-12-03260-f003:**
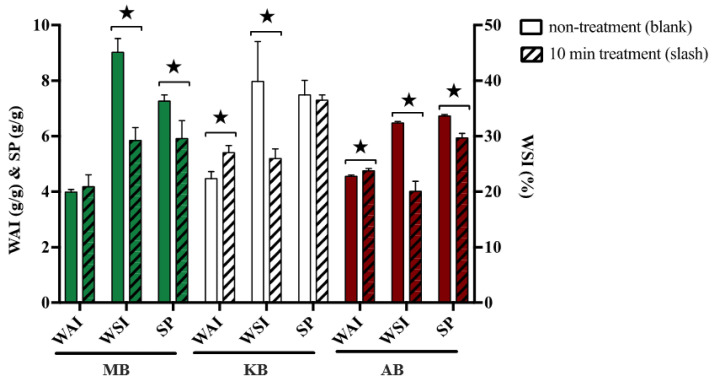
Effects of DBD-ACP treatment on hydration characteristics of different legumes. WAI means water absorption index, WSI means water solubility index, SP means swelling power, MB means mung bean, KB means kidney bean, and AB means adzuki bean. ★ In the same row of same kind of beans indicate significant differences at *p* < 0.05.

**Figure 4 foods-12-03260-f004:**
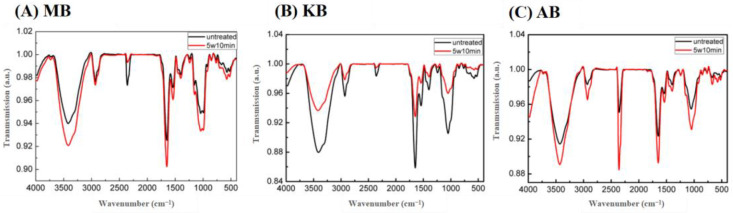
Effects of DBD-ACP treatment FI-TR of different legumes. MB means mung bean, KB means kidney bean, and AB means adzuki bean.

**Figure 5 foods-12-03260-f005:**
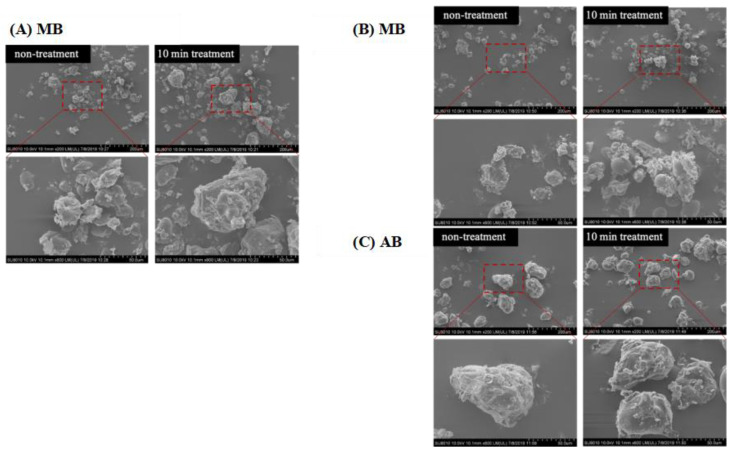
Effects of DBD-ACP treatment SEM of different legumes. MB means mung bean, KB means kidney bean, and AB means adzuki bean.

**Table 1 foods-12-03260-t001:** Effects of DBD-ACP treatment on chemical composition of three kinds of legumes.

	MB		KB		AB	
	Non-Treatment	10 Min Treatment	Non-Treatment	10 Min Treatment	Non-Treatment	10 Min Treatment
Carbohydrate (g/100 g)	63.83 ± 3.1	64.77 ± 2.57	65.71 ± 2.58	65.62 ± 1.95	68.41 ± 2.95	68.25 ± 1.69
Protein (g/100 g)	27.49 ± 1.87	28.09 ± 1.57	24.51 ± 1.95	25.12 ± 2.40	22.47 ± 1.90	22.49 ± 1.98
Fat (g/100 g)	2.51 ± 0.11	2.61 ± 0.16	1.71 ± 0.15	1.68 ± 0.09	1.49 ± 0.12	1.55 ± 0.18
TDF (g/100 g)	19.03 ± 0.41	20.19 ± 0.55 *	31.25 ± 1.09	31.73 ± 0.96	22.01 ± 0.58	23.15 ± 0.64 *
IDF (g/100 g)	15.82 ± 0.39	16.08 ± 0.51	27.1 ± 0.30	24.50 ± 0.41 *	17.86 ± 0.29	17.68 ± 0.30
SDF (g/100 g)	2.21 ± 0.40	4.11 ± 0.53 *	4.15 ± 0.02	7.23 ± 0.20 *	4.15 ± 0.36	5.47 ± 0.43 *

Results are expressed as mean ± standard error, expressed on dry matter basis. * In the same row of same kind of beans indicate significant differences at *p* < 0.05.TDF, total dietary fiber; IDF, insoluble dietary fiber; SDF, soluble dietary fiber; MB, mung bean; KB, kidney bean; AB, adzuki bean.

**Table 2 foods-12-03260-t002:** Effects of DBD-ACP treatment on amino acid composition of three kinds of legumes.

	MB		KB		AB	
	Non-treatment	10 Min Treatment	Non-Treatment	10 Min Treatment	Non-Treatment	10 Min Treatment
Thr (g/100 g)	0.77 ± 0.01	0.68 ± 0.02 *	0.88 ± 0.04	0.88 ± 0.02	0.63 ± 0.03	0.62 ± 0.02
Val (g/100 g)	1.06 ± 0.03	1.27 ± 0.03 *	1.14 ± 0.06	1.20 ± 0.03	0.84 ± 0.02	0.82 ± 0.03
Met (g/100 g)	0.16 ± 0.01	0.12 ± 0.00 *	0.08 ± 0.00	0.11 ± 0.01 *	0.12 ± 0.01	0.18 ± 0.01 *
Lle (g/100 g)	0.86 ± 0.07	1.03 ± 0.02 *	0.94 ± 0.05	0.99 ± 0.04	0.67 ± 0.03	0.63 ± 0.02
Leu (g/100 g)	1.66 ± 0.03	1.70 ± 0.08	1.59 ± 0.07	1.64 ± 0.03	1.30 ± 0.08	1.26 ± 0.02
Phe (g/100 g)	1.23 ± 0.09	1.32 ± 0.06	1.12 ± 0.03	1.19 ± 0.05	0.98 ± 0.05	0.95 ± 0.02
Lys (g/100 g)	1.62 ± 0.07	1.61 ± 0.03	1.50 ± 0.06	1.48 ± 0.03	1.40 ± 0.06	1.32 ± 0.04
Trp (g/100 g)	0.14 ± 0.01	0.15 ± 0.01	0.21 ± 0.04	0.21 ± 0.03	0.14 ± 0.02	0.14 ± 0.01

Results are expressed as mean ± standard error, expressed on dry matter basis. * In the same row of same kind of beans indicate significant differences at *p* < 0.05. MB, mung bean; KB, kidney bean; AB, adzuki bean.

**Table 3 foods-12-03260-t003:** Effects of DBD-ACP treatment on fatty acids compositions (%) of three kinds of legumes.

Fatty Acids (%)	MB		KB		AB	
	Non-Treatment	10 Min Treatment	Non-Treatment	10 Min Treatment	Non-Treatment	10 Min Treatment
C3:0	0.62 ± 0.03	0.66 ± 0.04	0.21 ± 0.02	0.40 ± 0.03 *	1.55 ± 0.01	1.44 ± 0.02 *
C11:0	8.53 ± 0.06	10.54 ± 0.09 *	7.49 ± 0.07	10.40 ± 0.04 *	17.43 ± 0.09	23.55 ± 0.08 *
C14:0	0.33 ± 0.01	0.47 ± 0.03 *	0.35 ± 0.00	0.44 ± 0.03 *	ND	0.34 ± 0.03 *
C15:0	ND	ND	0.27 ± 0.02	0.16 ± 0.01 *	ND	ND
C16:0	25.75 ± 0.02	34.24 ± 0.04 *	17.69 ± 0.04	20.28 ± 0.04 *	30.63 ± 0.04	33.23 ± 0.06 *
C16:1	ND	ND	0.50 ± 0.03	0.59 ± 0.02 *	ND	ND
C17:0	0.27 ± 0.01	0.43 ± 0.00 *	0.20 ± 0.02	0.25 ± 0.03	ND	ND
C18:0	5.38 ± 0.04	7.83 ± 0.03 *	3.56 ± 0.05	4.28 ± 0.03 *	9.09 ± 0.08	7.99 ± 0.06 *
C18:1	6.45 ± 0.06	8.22 ± 0.06 *	6.77 ± 0.00	12.31 ± 0.01 *	0.49 ± 0.02	0.51 ± 0.01
C18:2	40.38 ± 0.09	22.46 ± 0.07 *	30.66 ± 0.11	23.09 ± 0.09 *	19.67 ± 0.13	18.92 ± 0.12 *
C18:3	11.75 ± 0.06	13.74 ± 0.02 *	32.30 ± 0.09	25.02 ± 0.09 *	11.90 ± 0.07	10.18 ± 0.04 *
C20:0	0.52 ± 0.04	1.41 ± 0.04 *	ND	1.76 ± 0.01 *	5.08 ± 0.04	ND
C22:0	ND	ND	ND	ND	3.26 ± 0.03	2.09 ± 0.03 *
C23:0	ND	ND	ND	ND	0.90 ± 0.02	1.74 ± 0.02 *
∑ SFA	41.42 ± 0.14	55.58 ± 0.13 *	29.77 ± 0.13	38.98 ± 0.14 *	67.94 ± 0.07	70.38 ± 0.04 *
∑ MFA	6.45 ± 0.01	8.22 ± 0.01 *	7.26 ± 0.03	12.90 ± 0.02 *	0.49 ± 0.03	0.51 ± 0.02
∑ PFA	52.13 ± 0.12	36.19 ± 0.13 *	62.96 ± 0.10	48.12 ± 0.09 *	31.57 ± 0.11	29.10 ± 0.12 *

Results are expressed as mean ± standard error, expressed on dry matter basis. * In the same row of same kind of beans indicate significant differences at *p* < 0.05. ND, not detected; SFA, saturated fatty acids; MFA, monounsaturated fatty acids; PFA, polyunsaturated fatty acids. MB, mung bean; KB, kidney bean; AB, adzuki bean.

**Table 4 foods-12-03260-t004:** Effects of DBD-ACP treatment on in vitro digestibility of starch of different legumes.

	MB		KB		AB	
	Non-Treatment	10 Min Treatment	Non-Treatment	10 Min Treatment	Non-Treatment	10 Min Treatment
RDS (%)	13.87 ± 1.12	16.72 ± 1.33 *	4.65 ± 1.42	13.72 ± 0.87 *	11.41 ± 2.02	12.85 ± 0.95
SDS (%)	18.52 ± 1.76	19.86 ± 2.01	8.63 ± 1.05	13.85 ± 1.56 *	14.24 ± 1.01	14.57 ± 1.57
RS (%)	67.61 ± 2.35	63.42 ± 1.86	86.72 ± 3.88	72.43 ± 2.89 *	74.35 ± 1.77	72.58 ± 2.13

Results are expressed as mean ± standard error, expressed on dry matter basis. * In the same row of same kind of beans indicate significant differences at *p* < 0.05. RDS, rapidly digestible starch; SDS, slowly digestible starch; RS, resistant starch; MB, mung bean; KB, kidney bean; AB, adzuki bean.

**Table 5 foods-12-03260-t005:** Effects of DBD-ACP treatment on thermal properties of DSC analysis of different legumes.

	MB		KB		AB	
	Non-Treatment	10 Min Treatment	Non-Treatment	10 Min Treatment	Non-Treatment	10 Min Treatment
To (°C)	80.86 ± 0.53	84.68 ± 0.46 *	74.31 ± 0.15	74.97 ± 0.17 *	69.10 ± 0.28	69.97 ± 0.22 *
Tp (°C)	89.91 ± 0.16	90.97 ± 0.47 *	81.38 ± 0.50	81.40 ± 0.20	78.23 ± 0.66	78.95 ± 0.55
Tc (°C)	93.32 ± 0.19	95.72 ± 0.42 *	86.84 ± 0.36	85.56 ± 0.22 *	82.65 ± 0.32	82.08 ± 0.61
∆H (J/g)	1.32 ± 0.23	1.28 ± 0.12	2.14 ± 0.09	1.66 ± 0.10 *	2.35 ± 0.12	2.28 ± 0.13

Results are expressed as mean ± standard error, expressed on dry matter basis. * In the same row of same kind of beans indicate significant differences at *p* < 0.05. To, onset temperature; Tp, peak temperature; Tc, concluding temperature; ΔH, enthalpy; MB, mung bean; KB, kidney bean; AB, adzuki bean.

## Data Availability

All data are provided in the manuscript.

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
