# Peer review of "The Effect of Atmospheric Dielectric Barrier Discharge Cold Plasma Treatment on the Nutritional and Physicochemical Characteristics of Various Legumes"

_foods, 2023, doi:10.3390/foods12173260_

Round 1

Reviewer 1 Report

The manuscript has some nice results, however there are several points need to be addressed as follows:

1. Introduction: It is well-know that legumes are important food source. Please explain why mung bean (MB), kidney bean (KB) and adzuki bean (AB), not others were selected in this study.

2. Introduction: More critical discussion on Dielectric-barrier discharge atmospheric cold plasma is needed, e.g. treatment conditions relating to this study.

3. Section 2.2: Please provide more detailed information on the seeds in terms of moisture content, raw or dried materials, pre-treatment condition, etc.

4. Section 2.2: It is interesting to know about the sample treatment condition such as material load (kg/m2), thickness, among others.

5. Section 2.2: Why only treatment time was investigated whereas the other factors such as the input power at 5 W and the gap between the electrodes at 0.5 cm were fixed. Are their their optimal conditions.

6. Section 3.1: Please explain the mechanism of DBD-ACP reducing LOX. This information is particularly important in this study.

7. As microorganisms present in legumes are one of the most important problems, why they are not considered. Please explain/ discuss further.

8. Please discuss elsewhere in the manuscript on how the DBD-ACP treatment can be developed under pilot or industrial scale for practical applications.

The English language is good and needs a minor revision. 

Reviewer 2 Report

In this study, researchers investigated the effects of applying dielectric-barrier discharge atmospheric cold plasma (DBD-ACP) to various types of legume flours, including mung bean, kidney bean, and adzuki bean flours. The primary focus was on lipoxygenase (LOX) activity, which is a major contributor to oxidative rancidity in legumes. Overall, this study demonstrated that the application of DBD-ACP to legume flours effectively reduced LOX activity, positively influenced nutritional components, altered starch digestion characteristics, and modified the hydration and thermal properties of the flours.

In my assessment, the manuscript exhibits a commendable level of detail and reliability, making it suitable for publication in its current state.

Author Response

Thank you very much for your recognition of our article.

Reviewer 3 Report

The manuscript describes the effect of atmospheric dielectric barrier discharge cold plasma treatment on the nutritional and physicochemical properties of mung bean, kidney bean, and adzuki bean. The topic is interesting; however, the manuscript is poorly discussed and is not acceptable in its present form.

Comments:

- Line 38: Please Explain what cold plasma is and what uses it has in the food industry. You can use the following manuscript: 10.1016/j.ijbiomac.2022.12.321

- Line 195: Explain how cold plasma changes the amount of amino acids? Why is methionine decreased in mung beans and increased in kidney bean and adzuki bean?

Line 209: Please add more discussion for the obtained results.

Line 230: Please explain how CP treatment affects the digestibility of starch. Also please explain the influence of cold plasma on the depolymerization, etching, etc.

- Please add more discussion for all of the experiments.

Round 2

Reviewer 3 Report

The manuscript is acceptable.